# Multivariate Gaussian Bayes classifier with limited data for segmentation of clean and contaminated regions in the small bowel capsule endoscopy images

Vahid Sadeghi[1], Alireza Mehridehnavi[1]*, Maryam Behdad[2], Alireza Vard[1], Mina Omrani[3], Mohsen Sharifi[4], Yasaman Sanahmadi[1], Niloufar Teyfouri[5,6]

1 Department of Bioelectrics and Biomedical Engineering, School of Advanced Technologies in Medicine, Isfahan University of Medical Sciences, Isfahan, Iran, 2 Department of Electrical Engineering, Yazd University, Yazd, Iran, 3 Department of Mathematics and Computer Science, Amirkabir University of Technology, Tehran, Iran, 4 Gastroenterologist and Hepatologist Fellowship of Endosonography, Isfahan University of Medical Sciences, Isfahan, Iran, 5 Cancer Prevention Research Center, Isfahan University of Medical Sciences, Isfahan, Iran, 6 Omid Hospital, Isfahan University of Medical Sciences, Isfahan, Iran

* mehri@med.mui.ac.ir

**Data availability statement:** The code and CECleanliness dataset (including images and their corresponding binary masks) supporting this study are openly available on GitHub at the following link: [https://github.com/

## Abstract

A considerable amount of undesirable factors in the wireless capsule endoscopy (WCE) procedure hinder the proper visualization of the small bowel and take gastroenterologists more time to review. Objective quantitative assessment of different bowel preparation paradigms and saving the physician reviewing time motivated us to present an automatic low-cost statistical model for automatically segmenting of clean and contaminated regions in the WCE images. In the model construction phase, only 20 manually pixel-labeled images have been used from the normal and reduced mucosal view classes of the Kvasir capsule endoscopy dataset. In addition to calculating prior probability, two different probabilistic tri-variate Gaussian distribution models (GDMs) with unique mean vectors and covariance matrices have been fitted to the concatenated RGB color pixel intensity values of clean and contaminated regions separately. Applying the Bayes rule, the membership probability of every pixel of the input test image to each of the two classes is evaluated. The robustness has been evaluated using 5 trials; in each round, from the total number of 2000 randomly selected images, 20 and 1980 images have been used for model construction and evaluation modes, respectively. Our experimental results indicate that accuracy, precision, specificity, sensitivity, area under the receiver operating characteristic curve (AUROC), dice similarity coefficient (DSC), and intersection over union (IOU) are $0.89 \pm 0.07$, $0.91 \pm 0.07$, $0.73 \pm 0.20$, $0.90 \pm 0.12$, $0.92 \pm 0.06$, $0.92 \pm 0.05$ and $0.86 \pm 0.09$, respectively. The presented scheme is easy to deploy for objectively assessing small bowel cleansing score, comparing different bowel preparation paradigms, and decreasing the inspection time. The results from the SEE-AI project dataset and CECleanliness database proved that the proposed scheme has good adaptability.

vahidsadeghi97/Clean-and-contaminatd-region-segmentation-in-the-WCE-images-using-GBC-model]. Due to storage limitations on GitHub, additional datasets used in this study—including the Kvasir capsule endoscopy dataset, the SEE-AI project database, and the degraded Kvasir capsule endoscopy database—are publicly accessible on Figshare data repository with DOI: [https://doi.org/10.6084/m9.figshare.27645021.v1].

**Funding:** The author(s) received no specific funding for this work.

**Competing interests:** The authors have declared that no competing interests exist.

# 1. Introduction

Wireless capsule endoscopy (WCE), as a clinical diagnostic procedure, is capable of exploring the small bowel, the most complicated segment of the digestive system [1]. During an eight-hour journey of the WCE in the gastrointestinal (GI) tract, a large number of color frames are taken, which will later be reviewed by the specialist. The captured frames are not always clean because of food residues, turbid fluids, fecal materials, and air bubbles. These disturbing agents occlude the correct visualization of the bowel wall and consequently may lead to misinterpretation. Unfortunately, unlike duodenofibroscopy or ileocolonoscopy, the available WCE setup has not been equipped with suction or cleaning options [2]. By taking into account the mentioned limitations, different bowel preparation instructions such as a clear liquid diet, fasting overnight, using lavage and anti-foaming agents with different dosage and timing have been proposed to clean out the digestive tract prior to small bowel capsule endoscopy (SBCE) and subsequently improving the visualization quality of the captured video [3–7]. Despite the designed bowel preparation guidelines, choosing the optimal one is controversial because of the lack of objective scoring [8].

Small bowel visualization quality (SBVQ) assessment or investigating the effect of a wide variety of bowel preparation regimens in enhancing the captured images utilizing qualitative or quantitative scores [9] is a subjective and labor-intensive task that needs much preciseness which ends up in low to moderate inter- and intra-observer reproducibility [10]. So, there is still room for further study and enhancing the SBVQ score prediction.

In contrast to subjective human-based quality evaluation, objective computer vision-based algorithms can be utilized to assess the cleanliness of bowel images [11–13].

Generally, in developing computer vision-based techniques for clean and contaminated region segmentation in the WCE images, endoscopists must annotate a set of image data as training images in advance. In clinical applications, this process puts much burden on gastroenterologists.

In the current study, with the aid of a limited training samples (20 images), the gastroenterologist's expertise about the clean and contaminated regions will be encoded to a computer program. The prior probability of clean and contaminated regions will be computed. The pixel intensity distribution of clean and contaminated areas will be modeled with the help of probabilistic models. Finally, the probability of each pixel belonging to clean or contaminated classes will be calculated by applying the Bayes rule. Our color intensity-based approach can quantify how much of the frame is obstructed by small bowel contents.

The main contributions of this paper can be summarized as follows:

- A novel statistical scheme based on the visual color characteristics has been presented to detect the clean and contaminated regions on the pixel scale which can be utilized as an embedded lightweight model for SBVQ assessment.

- In contrast to prior research necessitating extensive image data, a pipeline will be developed with a few annotated images in the model construction phase.

- The robustness of the presented scheme against different real-world degradation scenarios that are available during the WCE imaging process will be evaluated.

The rest of this study has been structured as follows: in the first section, the related works in the clean and contaminated regions analysis of the WCE images will be reviewed. The materials and methods section includes four subsections. Firstly, three different datasets will be explored, which have been used to fit the Gaussian Bayes classifier (GBC) model, its performance evaluation, and transferability purposes. In the following part, the process of images selection for model construction and its assessment will be described. The next subsection

explains the GBC model construction and evaluation modes. The experiments section includes the procedures for providing the corrupted images from the original frames. In the results section, the clinical performance of the proposed scheme on the original images of three different datasets and the synthesized downgraded images will be reported, and eventually, the last sections will include discussion and conclusion.

## 2.  Related works

This section will briefly review the works related to the SBVQ assessment process. Most bubble-contaminated frame detection techniques consider the unique properties of the geometry of the bubbles. The specific round-shape characteristics of bubbles in the WCE images have been captured by using Gabor filter bank, ring shape selective (RSS) filters, grey-level of co-occurrence matrix (GLCM), fractal dimension, circular Hough transform and speeded-up robust features (SURF) saliency point detectors to assess air bubbles abundance in the SBCE frames [14–16]. In another study, for quantifying the amount of bubbles in the WCE images, low-pass filtering in the Fourier domain and suppressing the approximation coefficients in the 2-D discrete wavelet transform have been used [17]. Reviewed papers focused on detecting just only bubbles as one specific contaminating factor in the WCE images.

The ratio of mean color intensity values of the red to green (r/g) channels in the WCE images has been used as a cleanliness indicator [18]. The red-to-green ratio of 1.6 achieves satisfactory classification results in separating the adequately cleaned frames from inadequate ones [19]. Since the color tonality of the GI tract may change in different subjects and even in different gut segments, applying a fixed threshold ratio cannot be considered a global cleanliness measure.

To quantify the visible regions in the WCE images at first the illumination correction has been carried out. Subsequently, each pixel has been indexed as visible or non-visible using optimized thresholding on different color channels in the RGB color space [20]. The limitation of the mentioned study, which can be considered, is the lack of data diversity because they evaluated their technique on the images captured from only three patients.

In contrast to the mentioned single-step techniques, some researchers concentrated on detecting bubbles and turbid fluids in two cascade stages using color and texture attributes. In the first phase, the contaminated frames have been separated by extracting color attributes. Meanwhile, in the second phase, by exploiting the texture characteristics, the bubble-contaminated frames have been detected [21–23]. Due to the lack of pixel-based ground truth annotation masks, these algorithms work at the image level.

Some deep convolutional neural network (DCNN) architectures have been designed to assess the degree of cleanliness in the WCE frames. The assessment of small bowel cleanliness has been turned into a two- or five-class class classification problem [24,25]. However, acceptable results have been achieved, but these architectures in the training stage put a heavy burden of image labeling on physicians.

Other approaches may be closer to the current study in the context of clean and contaminated region segmentation. A novel DCNN architecture has been trained to distinguish the clean and contaminated patches in the WCE images automatically [26]. There are better solutions than regular rectangular patch selection in the mentioned study since each unique patch may include both clean and contaminated regions at the same time. Three well-known segmentation DCNN structures, DeepLab v3, FCN, and U-Net, have been trained to quantify mucosa cleanliness. Since the proposed DCCN structures are not optimal, a massive number of frames have been used to train them [27]. DCNN structures typically rely on extensive WCE-annotated image datasets for effective segmentation of clean and contaminated regions. The train data preparation procedure demands a significant amount of expert time and financial resources.

Deploying large DCNN models on resource-constrained devices, such as handheld diagnostic tools (mobile or portable diagnostic devices), may be impractical due to memory limitation and processing power. Not every one of the mentioned papers has evaluated their method on other datasets for generalization purposes. The robustness of the preceding studies against common real-world scenarios that can happen in the WCE imaging process has not been assessed.

We aim to design a novel rule-based scheme for the segmentation of clean and contaminated regions in the WCE images with a limited number of training samples (20 images) with minimum workload on the clinician.

## 3. Materials and methods

The block diagram of the clean and contaminated region localization scheme has been presented in Fig 1. Firstly, the downloaded images from three different databases have been annotated under the gastroenterologist's supervision. In the model construction phase, a limited number of pixel-based labelled images have been selected with the aid of 3-D color histogram and K-means clustering algorithm. Then the prior probability of the clean and contaminated classes has been calculated. Two distinct tri-variate GDMs have been fitted on clean and contaminated regions' concatenated color pixel intensity values. In the model assessment stage, each pixel in the test subset images is subjected to the two fitted probabilistic models to extract the clean and contaminated sections in each individual image.

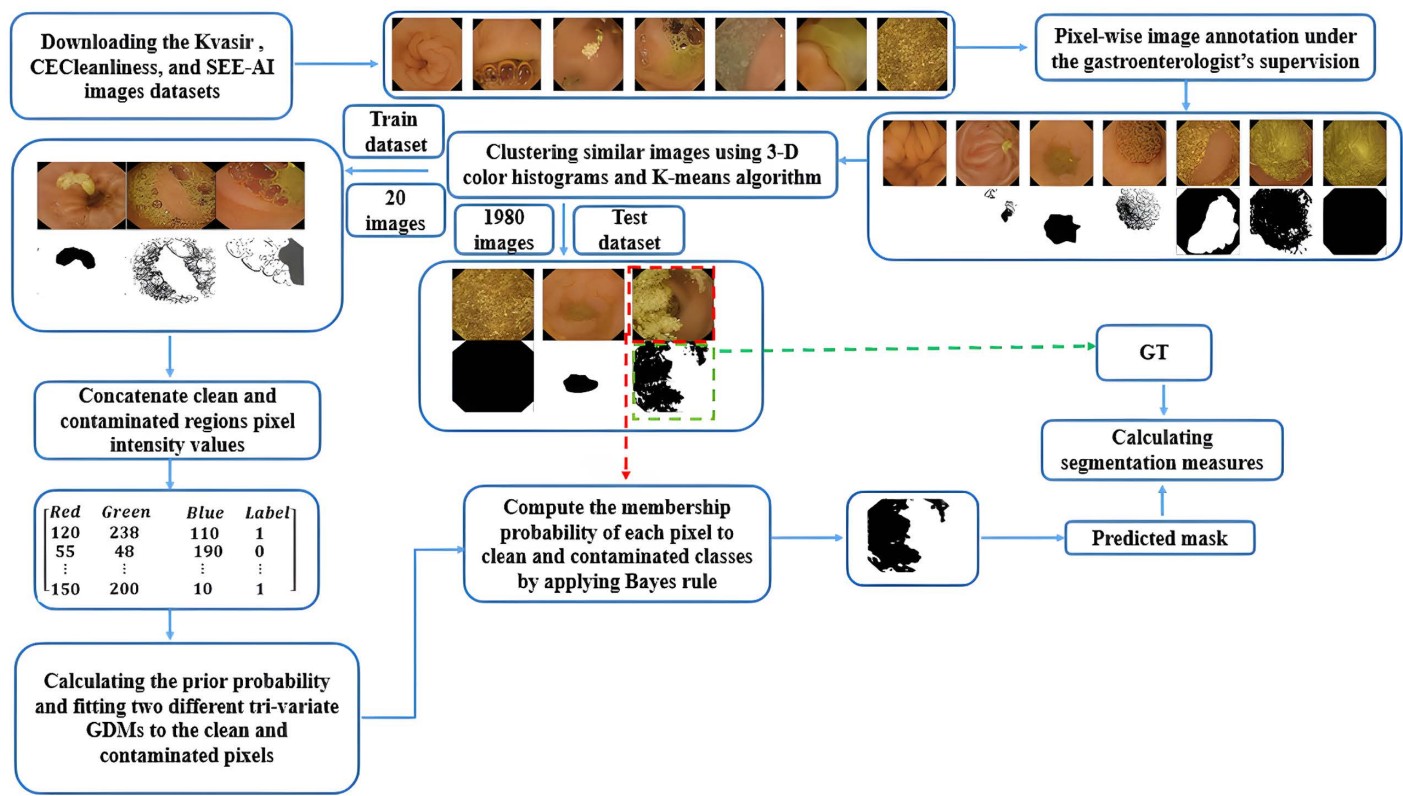

**Fig 1. Structure of the proposed clean and contaminated region segmentation method.**

## 3.1 Datasets

Three sets of SBCE frames, Kvasir [28], SEE-AI [29], and CECleanliness [26] capsule endoscopy datasets, have been selected for model construction and its performance evaluation on assessing the SBVQ score. Since the mentioned datasets have not been annotated at the pixel level, clean and contaminated regions have been manually segmented under the supervision of an experienced gastroenterologist. A binary mask has been created for each image in our annotated datasets. The values of 1 (white) or 0 (black) have been assigned to the clean and contaminated pixels, respectively. These manually created binary masks will subsequently be used in model construction and performance evaluation parts. The ground truth (GT) masks precisely partition the images into clean and contaminated regions.

**3.1.1 Kvasir capsule endoscopy dataset.** The WCE videos have been gathered from different clinical procedures carried out at the Department of Medicine, Bærum Hospital, Vestre Viken Hospital Trust in Norway from February 2016 to January 2018 utilizing the Olympus Endocapsule 10 system. Metadata have been intentionally deleted and files have been renamed with randomly generated names to guarantee full anonymization of image and video data in the Kvasir capsule endoscopy dataset, based on the Privacy Data Protection Authority approval and guidelines from the Regional Committee for Medical and Health Research Ethics - South East Norway. The Norwegian Privacy Data Protection Authority allowed the transport of anonymous images for the construction of the Kvasir capsule endoscopy database without participant consent form, exempting it from approval by the Regional Committee for Medical and Health Research Ethics - South East Norway. Due to anonymization and removal of metadata, the dataset is publicly shareable under Norwegian and General Data Protection Regulation (GDPR) laws. The Kvasir capsule endoscopy dataset is licensed under a Creative Commons Attribution 4.0 International (CC BY 4.0) License, allowing use, sharing, adaptation, distribution, and reproduction with appropriate credit to the original authors and source.

The Kvasir database images have been stored in the PNG format with a resolution of 336 pixels × 336 pixels. The Kvasir capsule endoscopy database contains 2906 and 34833 images for reduced mucosal view and normal classes, respectively. Since manually annotating the clean and contaminated regions in each image is time-consuming, 1939 and 61 images from the reduced mucosal view and normal categories, respectively, have been randomly selected for the manual pixel-level annotation stage. Randomly selected images include contaminating agents such as bubbles, bile, and food residue with a wide range of colors and different amounts of occupation.

Typical examples of annotated frames from the Kvasir capsule endoscopy dataset with different levels of contamination have been displayed in Fig 2. The first and second rows in Fig 2 represent the original frames and their corresponding manually annotated GT masks.

**3.1.2 SEE-AI project dataset and CECleanliness database.** For generalization purposes and robustness evaluation of the lightweight GBC model against the variations due to the intrinsic camera configurations, the SEE-AI project dataset and CECleanliness database have been utilized.

The SEE-AI project database collected anonymized videos from 954 patients who underwent SBCE (PillCam SB 3; Medtronic, Minneapolis, MN, USA) at Kyushu University Hospital between September 2014 and June 2021. Participants were able to choose to opt out of the study, approved by the Ethics Review Committee of Kyushu University (Approval No. 2021–213). Individual consent forms were not required as the SEE-AI project database is retrospective with anonymized data approved by the Ethics Committee. The SEE-AI project database is licensed under a Creative Commons Attribution 4.0 International (CC BY 4.0) License, allowing for copying, redistribution, combination, transfer, and modification with appropriate credit.

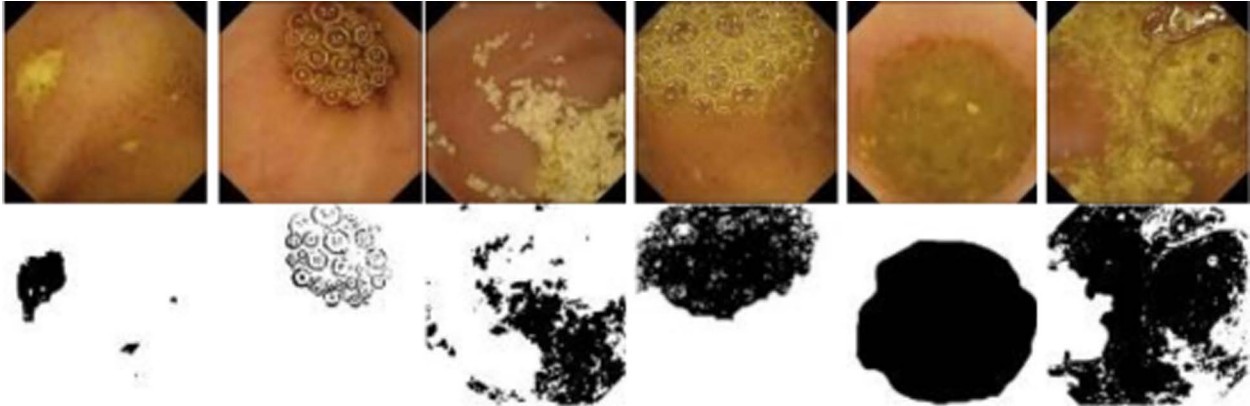

**Fig 2. Typical example of WCE images with their corresponding binary GT masks.** First and second rows: original images and their manually annotated binary masks, respectively.

The CECleanliness database was gathered at Hospital Universitari i Politècnic La Fe from Valencia with prior approval from its Ethics Committee of Clinical Research and written informed consent from all participants. The dataset curation process adhered to Spanish law for maintaining subject privacy.

The two mentioned datasets are a collection of small bowel capsule endoscopy images obtained using the PillCam™ SB 3. In the case of the SEE-AI project dataset, from a total of 6160 images of a normal class, 500 images have been randomly selected, and the majority of them (459 images) are contaminated. The number of 200 images from the 30 capsule endoscopy videos from the total number of 854 frames have been randomly chosen. These images, with a resolution of 576 pixels × 576 pixels, have been covered with different amounts of contaminating factors. Selected images from the SEE-AI and CECleanliness have been annotated in the same manner as the Kvasir capsule endoscopy dataset.

The authors of the Kvasir, CECleanliness, and SEE-AI project datasets acclaimed that ethical approval has been obtained from the Regional Committee for Medical and Health Research Ethics - South East Norway, Spanish law, and Review Committee of Kyushu University ("Approval No. 2021–213") respectively.

### 3.2 Grouping similar images using 3-D color histograms and K-means clustering

In constructing the training dataset for the GBC model, we implemented a systematic selection process aimed at ensuring a diverse and representative sample of both clean and contaminated regions.

Annotated images from the K-vasir capsule endoscopy dataset have been transformed from the RGB color space to the CIELAB color space due to the advantages of better color differentiation and perceptual uniformity in this color space. Subsequently, a 3-D color histogram has been constructed, where the three dimensions correspond to the three color channels (L, a, and b) of the CIELAB color space. Each of the three channels has been divided into a fixed number of bins (8 bins), resulting in a total of 512 bins. The 3-D color distribution of each image has been described by counting the number of pixels that fall into each individual bin. Each image has been represented as a 1-D feature vector, obtained by flattening the 3-D color histogram.

We conducted an initial analysis of color distributions within the entire dataset to identify patterns indicative of clean and contaminated regions. This involved clustering techniques to

categorize images based on their color characteristics, which helped us identify groups that included a variety of conditions.

To group similar images based on their color distributions, the K-means clustering algorithm has been applied to the resulting feature vectors. The K value (number of clusters) has been set to 20, meaning all of the images have been clustered into twenty groups. It would be better to select images that best represented the pixel intensity distributions of each class (clean and contaminated) within their respective clusters. From each cluster, one image has been randomly selected. These 20 representative images from all of different clusters play the role of the model fitting subset. All of the remaining 1980 images have been considered as the evaluation subset. By following the mentioned procedure, we aimed to construct a training set that captures the inherent variability in real-world WCE images, thereby enhancing the model's generalizability and robustness across different datasets and clinical scenarios. The values for the number of bins in the 3-D color histogram construction and the K in the K-means clustering algorithm have been set experimentally.

### 3.3 Model construction and evaluation phases

Clean visible mucosa usually appears pink, orange, and light brown. The obscured mucosa by luminal content or bubbles manifests themselves in a wide range of colors such as green, yellow, or dark brown. Since clean and contaminated regions in the WCE images manifest themselves with different characteristics in terms of their color patterns, color is the first clue to segregating clean regions from contaminated ones.

We are going to encode the expected color characteristics of the clean and contaminated regions as prior knowledge of the expert into the statistical model to guide the segmentation process to accurately differentiate between clean and contaminated regions in the WCE images.

Although there are many color feature descriptors, such as color moments, color layout, color coherence vector, and color histogram, the last one is the most convenient to describe its distribution.

From now on $x = [r, g, b]$ as observation is a three-dimensional vector, the pixel intensity values of the red ($r$), green ($g$), and blue ($b$) components in the RGB color space. The $y$ is what we want to predict: the input pixel is clean or contaminated. The $k$ refers to the number of classes, which equals two as clean and contaminated.

In other words, for a given pixel ($x$), it is desired to estimate the $p(y = clean|x)$, and $p(y = contaminated | x)$. The $x$ will be assigned to a class with higher probability. Here, the Bayes rule comes into play, which can help estimate the $p(y = clean | x)$, and $p(y = contaminated | x)$ for pixel classification. To apply the Bayes rule in the context of clean and contaminated region segmentation, pixel classification, prior probability, and two probabilistic models must be established, two distributions from which the clean and contaminated pixels are assumed to be obtained as two random samples.

Bayes rule is essentially a way to reverse a conditional probability; suppose we know the $p(x | y = k)$, but we want to find $p(y = k | x)$. In other words, by using Bayes rule probability of being clean or contaminated can be calculated on the base of pixel color intensity values.

The idea is to calculate the $p(y = k | x)$ and pick the $k$ with maximum probability.

$$k^* = \arg\max_k p(y = k | x) \tag{1}$$

In Eq (1), $p(y = k | x)$ is known as posterior probability. By applying the Bayes rule, it can be rewritten as:

$$k^* = \arg\max_k \left[ \frac{p(x|y=k)\,p(y=k)}{p(x)} \right] \tag{2}$$

Where in Eq ($2$), $p(x\,|\,y=k)$, $p(x)$, and $p(y=k)$ are known as likelihood, evidence, and prior probability respectively.

Since $p(x)$ is the same for each $k$ so, we can effectively drop it because it is constant concerning $k$. Therefore the Eq ($1$) leads to:

$$k^* = \arg\max_k \ p(x|y=k)\,p(y=k) \tag{3}$$

All terms in the last equation can be estimated from the pixel-level annotated training image dataset, the $pp(y=k)$ will be calculated, and the probability distribution of $p(x|y=k)$ must be modeled. To this end and to capture the color patterns present in the clean and contaminated regions, 20 frames have been chosen (one random image from each individual cluster) from the total number of 2000 images. Thanks to the manually annotated GT masks, the color pixel intensity values of the clean and contaminated regions from the selected images have been concatenated together separately. The concatenated RGB pixel intensity values for each of the clean and contaminated regions is a $N_k$ by $D$ matrix:

$$X_k = \begin{bmatrix} R_k^1 & G_k^1 & B_k^1 \\ \vdots & \vdots & \vdots \\ R_k^{N_k} & G_k^{N_k} & B_k^{N_k} \end{bmatrix} \tag{4}$$

Where $N_k$ is the number of observations (pixels) from the class $k$, and $D$ is the number of input features ($D = 3$).

The prior probability can be computed as follows:

$$p(y=k) = \frac{number\ of\ training\ pixels\ belonging\ to\ class\ k}{total\ number\ of\ training\ pixels} \tag{5}$$

The $p(x\,|\,y=k)$ depends on the distribution of the color pixel intensity values. In Fig 3, along with the empirical histogram, the Gaussian probability distribution function (PDF) has been fitted to the concatenated RGB pixel intensity values of clean and contaminated regions. As shown in Fig 3, given $y$ targeted values, the distribution of each of the $x$ input variables looks roughly like one-dimensional Gaussian distribution for both clean and contaminated classes. Since there is a correlation between RGB color component values as input features, the $p(x|y = k)$ can be modeled by the joint multivariate Gaussian distribution which means first (mean) and second order statistics (covariance matrix) can be enough to characterize them.

$$p(x|y=k) = N(x; \mu_k, \Sigma_k) = \frac{1}{\sqrt{(2\pi)^D |\Sigma_k|}} exp\left(-\frac{1}{2}(x-\mu_k)^T \Sigma_k^{-1} (x-\mu_k)\right) \tag{6}$$

Where $\mu_k$ and $\Sigma_k$ are the $1 \times 3$ mean vector and $3 \times 3$ symmetric covariance matrix of class $k$ respectively. The $|\,.\,|$ denotes the determinant operator. Each class of data points has its own distribution.

The $\mu_k$ and $\Sigma_k$ can be estimated with respect to the prior knowledge of clean and contaminated pixel-level annotated frames as follows:

$$\widehat{\mu_k} = E(X_k) = \left[\mu_k^R, \mu_k^G, \mu_k^B\right] \tag{7}$$

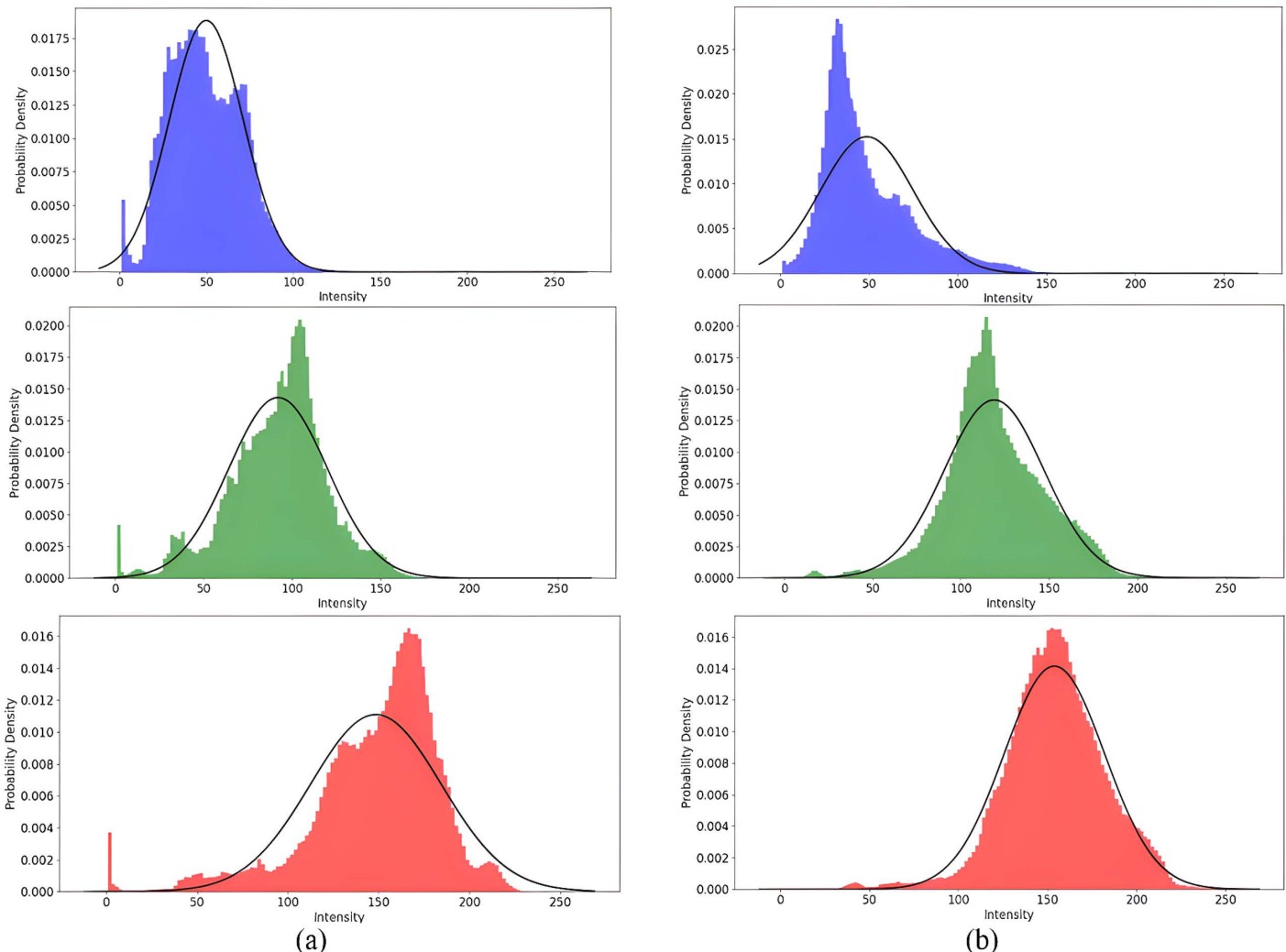

**Fig 3. Demonstration of the distributions and model-fitting performance on the concatenated RGB pixel intensity values of (a): clean and (b): contaminated regions.** In the first, second, and third rows, the empirical histogram and the fitted Gaussian distribution models have been plotted in the R, G, and B color components, respectively (as their color indicates).

$$\widehat{\Sigma_k} = Cov(X_k) = E\left(\left(X_k - \widehat{\mu_k}\right)^T \left(X_k - \widehat{\mu_k}\right)\right) = \begin{bmatrix} \sigma_k^{RR} & \sigma_k^{RG} & \sigma_k^{RB} \\ \sigma_k^{GR} & \sigma_k^{GG} & \sigma_k^{GB} \\ \sigma_k^{BR} & \sigma_k^{BG} & \sigma_k^{BB} \end{bmatrix} \tag{8}$$

Where $(.)$ and $Cov(.)$ are expected value and covariance operator, respectively.

A number will be returned by finding the prior probability and conditional probability parameters and plugging the $x$, the color intensity value of the test image for each class. The $k$ with the highest posterior probability will be picked. The $p(y = k)$ is constant, and the term $p(x|y = k)$ is variable because it depends on the input features. After completing the model construction part, we can extract the clean and contaminated pixels in the trial images. The presented scheme will be called the Gaussian Bayes classifier (GBC) because of using the Bayes rule and the assumption that $p(x|y = k)$ follows the Gaussian distribution.

### 3.4 Experiments

**3.4.1 Original images.** The 5 round trials have been used on each dataset to train and evaluate the low-complexity GBC model. In each round, from each group of similar images clustered by the help of 3-D color histogram and K-means clustering algorithm only one random image has been selected for model fitting, and the remaining images have been utilized in the inference phase. Eventually, the mean ($\pm$ std) of the obtained scores from the five trials are used to evaluate the model performance on different datasets.

**3.4.2 Corrupted images.** Not every one of the preceding studies has evaluated their method's stability against common degradations which are inevitable in the WCE image capturing procedure. With the aid of the Albumentation library [30], the number of 2000 WCE images have been processed to simulate real-world poor capturing condition in the small bowel, such as condensation, Gaussian noise, poor focus, uncontrollable movement of the capsule through the small intestine, non-uniform illumination, chromatic aberration. The robustness of the GBC model against each individual corruption will be assessed on the synthesized images.

The number of 6 diverse transformations such as fogging effect, adding Gaussian noise, defocus, motion blur, random brightness, and color jitter have been applied to 2000 images from the Kvasir capsule endoscopy dataset. Since these perturbations can occur during imaging process at varying levels of intensity, each corruption has different degree of severity.

3.4.2.1 **Fogging:** Moisture buildup in the capsule endoscopy, which is known as fogging, can occur due to environmental conditions such as high humidity or temperature changes. Moisture may accumulate on the lens or sensor, leading to frames with reduced clarity or hazy images. The lower and upper limits for the fog intensity coefficient have been set to 0.25 and 1. A value of 0.05 has been assigned to the transparency of the fog circles.

3.4.2.2 **Gaussian noise:** Gaussian noise is one of the common types of noise that is available in digital imaging and can degrade the quality of the captured images, making it more challenging to interpret the images accurately. Gaussian noise in the WCE images may be due to various sources such as embedded camera sensors, ambient lighting conditions, and the characteristics of the tissues being imaged. A Gaussian noise with zero mean and a variance in the range of (5, 300) has been added to each color channel of the the Kvasir baseline images.

3.4.2.3 **Defocus:** Defocus artifact is a significant challenge encountered in capsule endoscopy frames. This phenomenon arises from the inability of the embedded camera to focus sharply on the targeted area. Peristalsis physiological movements within the GI tract can cause the camera to shift position, leading to blurry or out-of-focus images. The capsule endoscopy usually has a fixed depth of focus up to 3 cm [31]. The currently available capsule endoscopy setup does not have the autofocus capability.

3.4.2.4 **Motion blur:** Motion blur in capsule endoscopy images can occur due to the passive movement of the capsule endoscope within the GI tract. As the capsule travels through the digestive system, it is subjected to various physiological movements, including naturally occurring peristalsis, respiration, and cardiac pulsations. These movements can cause the camera to shift position rapidly, resulting in blurred or distorted images [32].

A two-dimensional kernel has been designed and convolved with the original images to simulate the linear motion of the embedded camera in different directions with different speeds. The rotation angle spanned from 0° to 360° with a rotation step size equal to 15°. The movement speed has been randomly selected in the range of 1 to 25 pixels.

3.4.2.5 **Random brightness:** In the capsule endoscopy imaging process, the incident light is not evenly distributed across the entire field of view because of the presence of tissues with variability in their characteristics and surface topography.

Insufficient or excessive light intensity is very common in the WCE images, where the former and latter cause dark (shadowed) and hotspots (glare) areas, respectively. The illumination distribution of currently available capsule endoscopy is out of control [33].

The illumination of the baseline curated image has been intentionally changed using the power-law transformation. The transformation function can be expressed as below:

$$S = r^{\frac{1}{\gamma}} \tag{9}$$

Where $r$ is the normalized pixel intensity values of the original image and $S$ is the pixel values of the transformed image. Gamma ($\gamma$) values less than one will shift the image towards the darker end of the spectrum, while gamma greater than one will make the image appear brighter. In our experiments the gamma value has been changed randomly from 0.75 to 1.25.

3.4.2.6 **Color jitter:** Color inconsistency in the capsule endoscopy images has been simulated by random fluctuation in color intensity in the captured frames. The baseline images have been transformed from the original RGB to the HSV color space. Subsequently, the hue, saturation, and value of the WCE images have been randomly altered to simulate the shift in color. The range for random shifting of hue, saturation, and value has been set to 5.

## 4. Results

### 4.1 Evaluation of the performance of GBC on the Kvasir, CECleanliness, and SEE-AI datasets

In this section, the experimental results of the presented scheme on the clean and contaminated region localization have been presented. Quantitative assessment has been used for performance evaluation by comparing the clean and contaminated areas delineated manually under specialist supervision with those marked by our method outcome. For better generalization, the explained procedure has been repeated five times for each individual dataset, meaning at each time, 20 annotated images from each unique dataset have been selected as a model-fitting subset, and the rest of the images in each database have been used for model evaluation. The mean (±std) of accuracy, DSC, and IOU have been computed as segmentation indices for different trials on three independent database and reported in Table 1.

### 4.2 Assessment of the performance of the GBC model on the corrupted Kvasir capsule endoscopy dataset

Since the WCE is a visual diagnostic modality, the quality of captured images may affect the performance of the GBC model on the clean and contaminated regions segmentation. In this subsection the contribution of different degradation scenarios on the image visualization quality and subsequently on the segmentation of clean and contaminated regions by the GBC has been investigated.

We evaluated the SBVQ prediction of the GBC model against wide variety of corruptions that are common in the WCE imaging process. The GBC classifier has been fitted on the original Kvasir capsule endoscopy dataset images. Then the resultant fitted model has been evaluated on the algorithmically downgraded images. Table 2 gives a complete overview of the clean and contaminated regions segmentation of the GBC on the synthetized Kvasir capsule endoscopy images.

In Fig 5 the first to fourth rows illustrates the original images, degraded images, corresponding GT masks, and the outcomes of the GBC model, respectively. The type of corruption have been provided on the bottom side of Fig 5.

**Table 1. Performance of the presented scheme for the clean and contaminated region segmentation on three different databases.**

| Dataset<br>Measure | Kvasir | SEE-AI | CECleanliness |
|---|---|---|---|
| Accuracy | 0.89 ± 0.07 | 0.92 ± 0.07 | 0.88 ± 0.08 |
| Precision<br>(Positive Predictive Value) | 0.91 ± 0.07 | 0.92 ± 0.07 | 0.95 ± 0.04 |
| Specificity<br>(True Negative Rate) | 0.73 ± 0.20 | 0.71 ± 0.30 | 0.85 ± 0.17 |
| Sensitivity<br>(Recall or True Positive Rate) | 0.90 ± 0.12 | 0.97 ± 0.05 | 0.88 ± 0.11 |
| Area Under the Receiver Operating Characteristic Curve<br>(AUROC) | 0.92 ± 0.06 | 0.81 ± 0.20 | 0.79 ± 0.12 |
| DSC<br>(F1-Score) | 0.92 ± 0.05 | 0.94 ± 0.05 | 0.91 ± 0.06 |
| IOU | 0.86 ± 0.09 | 0.90 ± 0.08 | 0.84 ± 0.10 |

The performance of the GBC model in segmenting clear and contaminated regions within five randomly chosen images from the Kvasir, SEE-AI, and CECleanliness datasets has been illustrated graphically in Fig 4.

**Table 2. Performance of the GBC model for segmentation of clean and contaminated regions on degraded images.**

| Scenario<br>Measure | Original images | Fogging | Gaussian noise | Defocus | Motion blur | Random brightness | Color jitter |
|---|---|---|---|---|---|---|---|
| Accuracy | 0.89 ± 0.07 | 0.88 ± 0.06 | 0.86 ± 0.07 | 0.86 ± 0.09 | 0.87 ± 0.08 | 0.87 ± 0.07 | 0.86 ± 0.09 |
| Precision | 0.91 ± 0.07 | 0.90 ± 0.08 | 0.91 ± 0.08 | 0.90 ± 0.08 | 0.89 ± 0.08 | 0.90 ± 0.09 | 0.90 ± 0.09 |
| Specificity | 0.73 ± 0.20 | 0.72 ± 0.20 | 0.75 ± 0.19 | 0.71 ± 0.24 | 0.72 ± 0.20 | 0.70 ± 0.20 | 0.70 ± 0.29 |
| Sensitivity | 0.90 ± 0.12 | 0.92 ± 0.09 | 0.89 ± 0.11 | 0.90 ± 0.13 | 0.89 ± 0.11 | 0.90 ± 0.11 | 0.90 ± 0.13 |
| AUROC | 0.92 ± 0.06 | 0.91 ± 0.06 | 0.91 ± 0.06 | 0.92 ± 0.06 | 0.91 ± 0.06 | 0.91 ± 0.06 | 0.90 ± 0.10 |
| DSC | 0.92 ± 0.05 | 0.91 ± 0.06 | 0.89 ± 0.07 | 0.89 ± 0.08 | 0.89 ± 0.07 | 0.90 ± 0.07 | 0.89 ± 0.09 |
| IOU | 0.86 ± 0.09 | 0.83 ± 0.09 | 0.81 ± 0.11 | 0.81 ± 0.12 | 0.81 ± 0.12 | 0.82 ± 0.11 | 0.81 ± 0.13 |

For a qualitative comparison of the clean and contaminated region segmentation in degraded images from each of six individual degradation scenarios, one sample image along with its manually binary annotated GT and outcome mask of the GBC classifier has been depicted in Fig 5.

## 4.3 Comparison of the GBC to state-of-the-art DCNN segmentation architectures

The performance of the proposed scheme has been compared to the state-of-the-art DCNN architectures for clean and contaminated regions segmentation on the Kvasir capsule endoscopy dataset. The constructed dataset has been stochastically split with an 80:20 ratio for train and test subsets. It is worth to mentioning, however, the training and evaluation of U-Net and Pix2Pix DCNNs have been carried out on the Google Colaboratory virtual machine with an NVIDIA Tesla T4 GPU with 16 GB of VRAM (Video Random-Access Memory), the GBC model has been fitted and tested using an Intel Core i-5 CPU @ 2.50 GHz clock with 4.00 GB RAM. The value of different segmentation measures, the number of images in the train and test mode on the Kvasir capsule endoscopy dataset, along with the number of parameters for the U-Net, Pix2Pix, and the GBC models have been reported in Table 3.

## 4.4 Comparison of SBVQ between the specialist and the GBC model

In this subsection, we aim to compute the Pearson's correlation coefficient as the inter-observer agreement on bowel cleansing between the specialist and the GBC model. The

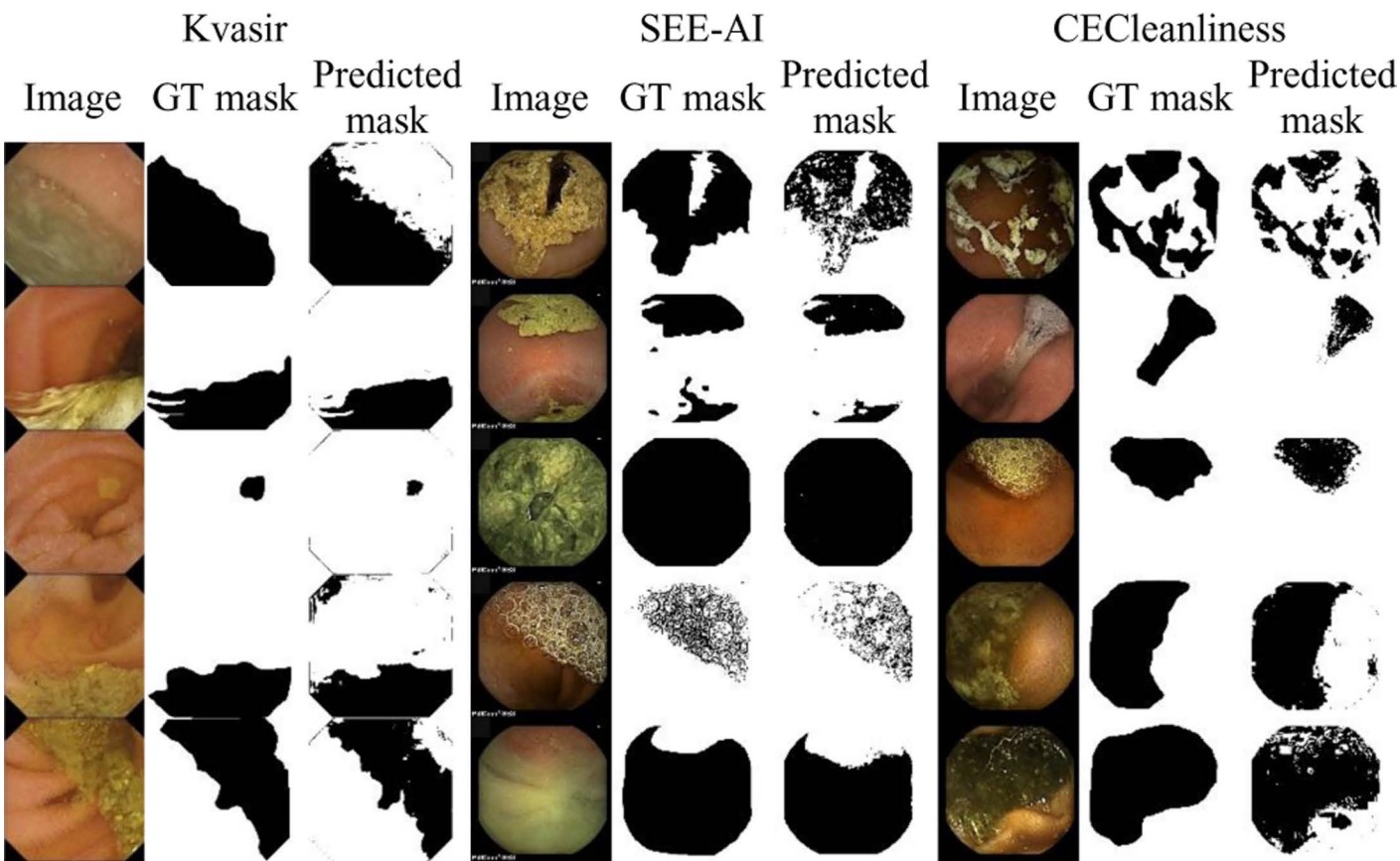

**Fig 4. Typical images from three different datasets along with their GT and predicted mask by the GBC model.**

scatterplot in Fig 7 shows that there is a high correlation for SBVQ prediction between the gastroenterologist and the GBC model (Pearson's correlation coefficient = 0.91). In other words, the GBC model can satisfactorily capture clean and contaminated regions at any level in the test images. It is worth to note that Fig 7 illustrates the relationship between the clean region in the image as evaluated by an experienced gastroenterologist (x-axis) and the predicted clean region as assessed by the GBC model (y-axis). The "clean region" is defined as the proportion of pixels in the image that are free from contaminants such as bubbles, debris, or other obstructions that might obscure the small bowel wall. In other words, it represents the ratio of clean pixels to the total number of pixels in the image, providing a quantitative measure of the image's visual clarity.

## 5. Discussion

In a typical WCE process, many captured frames are medically irrelevant, which means they contain mainly intestinal juices, bubbles, or debris. Therefore, finding unimportant regions of a frame that do not carry clinical information is beneficial to reducing video reading time and evaluating small bowel cleanliness. There are many qualitative and quantitative grading systems for SBVQ evaluation [9]. All these scoring systems rely on the technicians' subjective evaluation and depend on the viewers' notion of cleanliness [10]. The potential subjective opinion-related bias may result in a wide difference in grading the adequacy of bowel preparation among gastroenterologists.

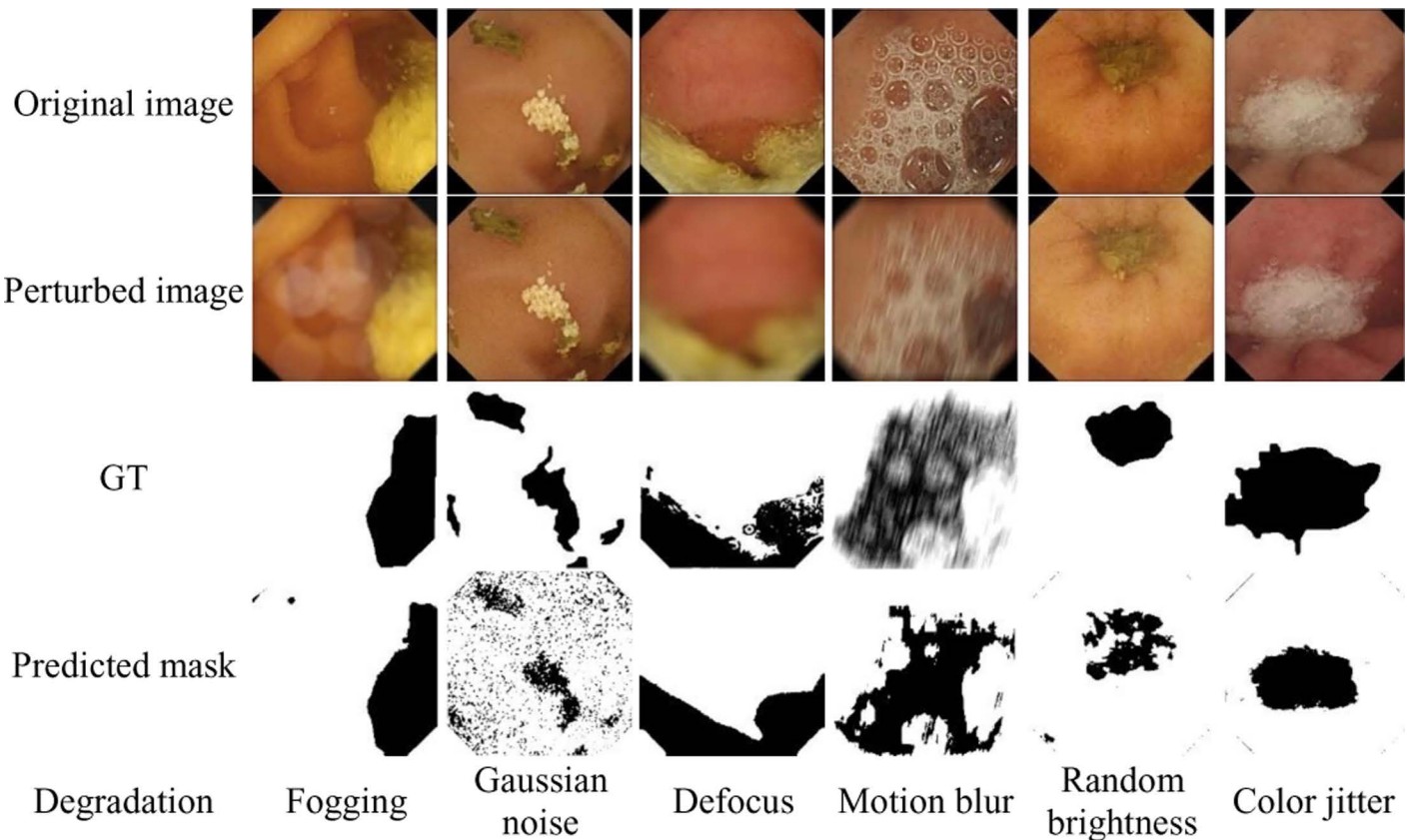

**Fig 5. The predicted black-and-white masks by the GBC model for some corrupted images subjected to different degradations.**

**Table 3. Comparison between the GBC model and the state-of-the-art DCNNs for clean and contaminated regions segmentation based on supervised learning in terms of different measures.**

| Model Measure | U-Net | Pix2Pix | GBC |
|---|---|---|---|
| Accuracy | 0.93 ± 0.02 | 0.94 ± 0.06 | 0.89 ± 0.07 |
| Precision | 0.94 ± 0.02 | 0.94 ± 0.07 | 0.91 ± 0.07 |
| Specificity | 0.74 ± 0.07 | 0.70 ± 0.30 | 0.73 ± 0.20 |
| Sensitivity | 0.97 ± 0.02 | 0.98 ± 0.03 | 0.90 ± 0.12 |
| AUROC | 0.97 ± 0.02 | 0.97 ± 0.08 | 0.92 ± 0.06 |
| DSC | 0.93 ± 0.07 | 0.96 ± 0.04 | 0.92 ± 0.05 |
| IOU | 0.89 ± 0.08 | 0.92 ± 0.08 | 0.86 ± 0.09 |
| Number of training images | 1600 | 1600 | 20 |
| Number of test images | 400 | 400 | 1980 |
| Number of parameters | 120,825 | 57,196,292 | 26 |
| Model training (construction) time(second) | 741 | 2782 | 25 |

For qualitative visualization of the clean and contaminated regions segmentation, a few randomly selected frames, along with their corresponding manually black-and-white annotated GT masks and the outcomes of U-Net, Pix2Pix, and GBC, have been depicted in Fig 6.

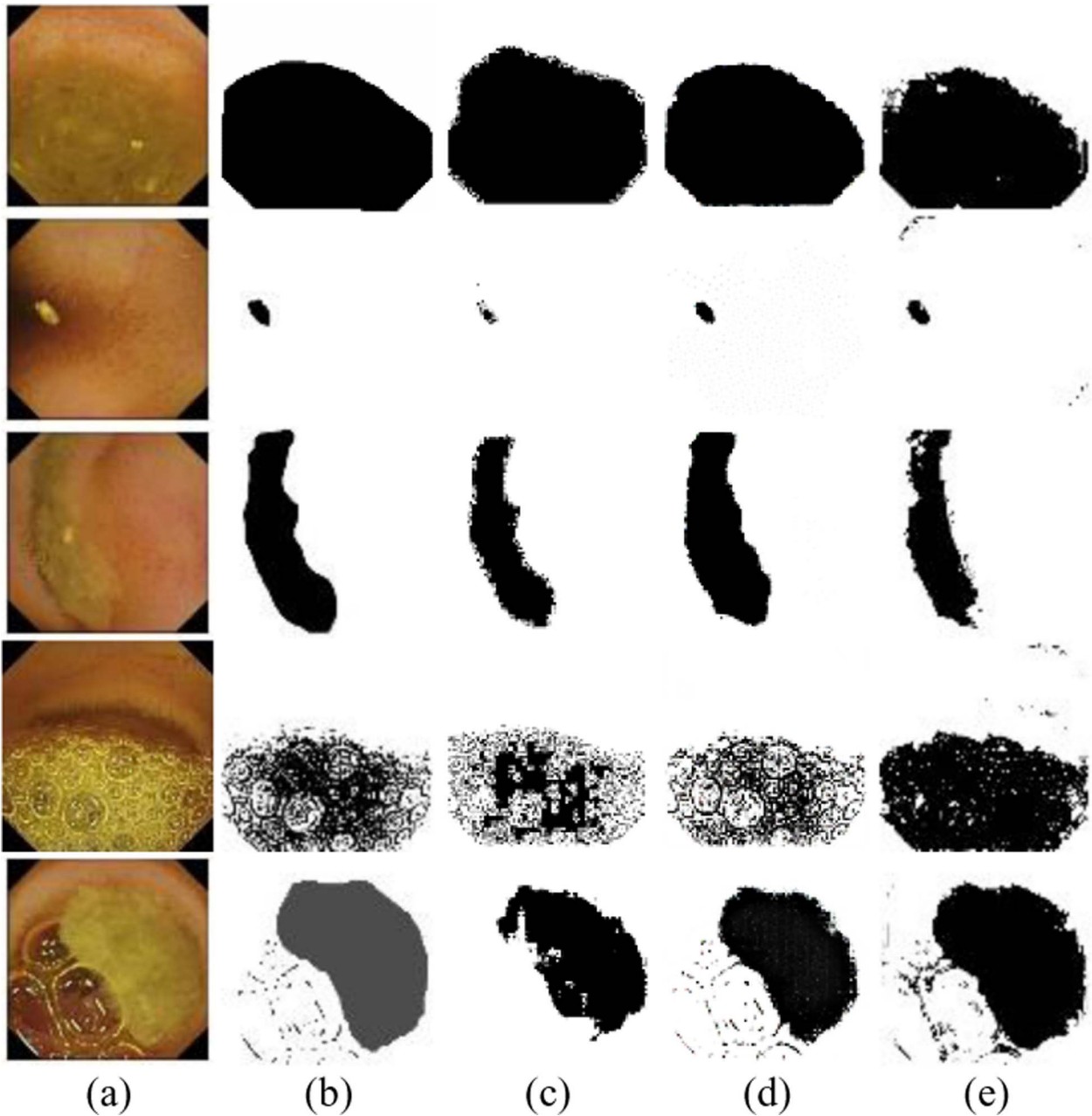

(a)        (b)        (c)        (d)        (e)

**Fig 6. The generated black and white segmentation mask along with the original image and their corresponding GT masks of some randomly chosen frames.** (a) Original images. (b) GT mask. (c) Output mask by the U-Net. (d) Predicted mask of Pix2Pix. (e) Outcome mask of the GBC.

The presented manuscript uses statistical modeling to assess SBVQ using objective segmentation of the mucosal regions occupied by interfering factors such as bile, debris, food residues, and bubbles. This statistical-based grading system can turn subjective evaluation into an objective assessment process, letting a lightweight computer program analyze the WCE video frames. By incorporating prior knowledge about the color pixel intensity values of the clean and contaminated areas, the GBC model can effectively capture the visual color characteristics of clean and contaminated regions.

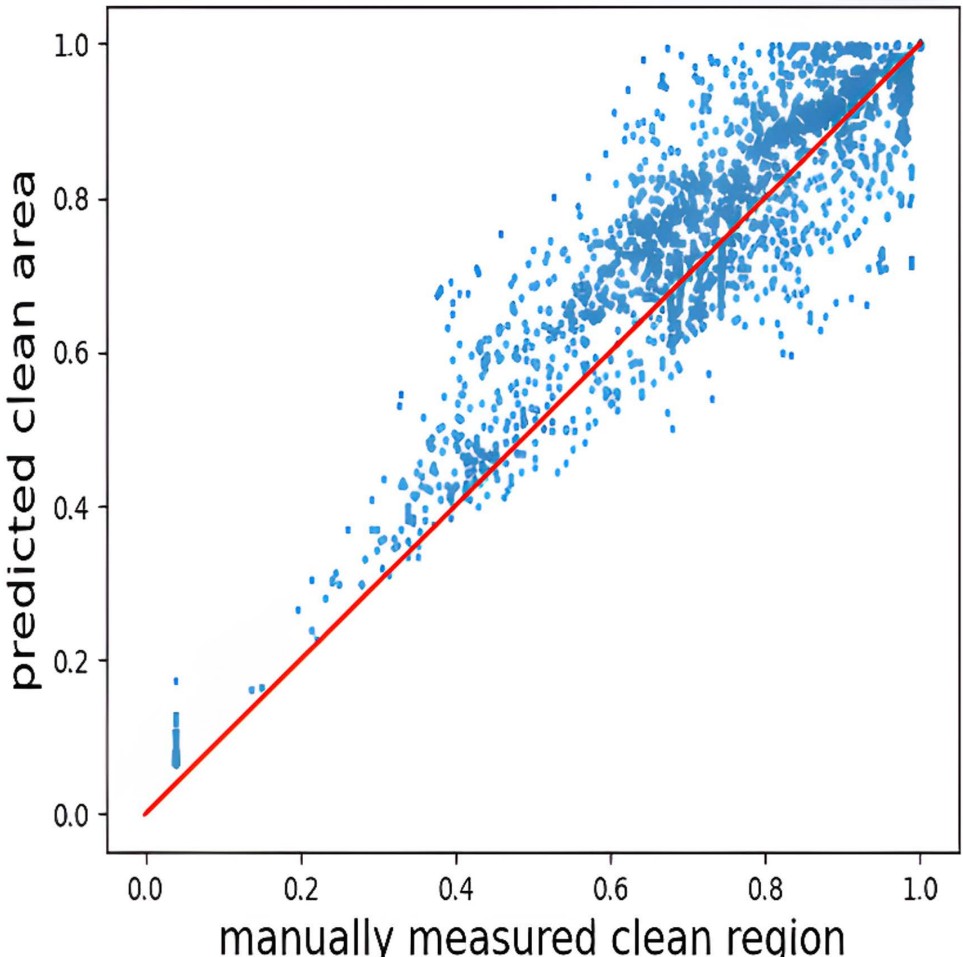

**Fig 7. Scatterplot demonstrates a comparison between the area of clean region prediction by the automated GBC segmentation model and the manually annotation for the test images.**

In the clustering phase, the dataset pool was divided into 20 groups based on color distributions. From each cluster, one image was stochastically selected. These 20 selected images, along with their corresponding GT masks, were then utilized in the GBC model construction phase. To assess the performance of the proposed paradigm, the process of grouping similar images, constructing the GBC model, and conducting evaluations was repeated five times for each curated database.

None of the reviewed studies for clean and contaminated region segmentation in the WCE images evaluated their presented algorithm on different datasets. The CECleanliness and SEE-AI project datasets, each presenting unique challenges, were utilized to assess the adaptability and robustness of the proposed scheme across various subjects, image capture conditions, and types of capsule endoscopy. By testing the model on these diverse datasets, we can better understand its ability to generalize across different clinical scenarios and hardware configurations, ensuring reliable performance in varying real-world conditions.

From Table 1, it can be inferred that the GBC model adapts very well to alternative datasets and reacts properly to new capturing conditions. Since the color diversity of the contaminated

patterns in the SEE-AI dataset is limited, its results are much better than the Kvasir capsule endoscopy dataset.

The Kvasir capsule endoscopy dataset contains images with varying degrees of contamination and diverse lighting conditions. The primary challenge here is the color variability caused by differences in bowel contents, which can impact segmentation accuracy. The GBC model adapted by leveraging prior knowledge of color intensity distributions to effectively distinguish between clean and contaminated regions.

The CECleanliness dataset presents a different challenge, characterized by high variability in illumination conditions. Variations in brightness during image capture complicate segmentation, as both clean and contaminated regions can appear under different lighting conditions. The GBC model addressed this by training Gaussian models on data with a wide range of lighting conditions. This allowed the model to learn the variance in pixel values caused by lighting differences, improving generalization during testing. By accounting for varying illumination, the model could focus on intrinsic features of the tissue or contamination, rather than being misled by brightness or shadow effects.

Both the SEE-AI Project dataset and the CECleanliness dataset differ in resolution compared to the Kvasir dataset. However, since the GBC model treats clean and contaminated region segmentation as a pixel classification task, the resolution of the captured images has minimal impact. This ability to classify on a pixel level enables the GBC model to generalize well across datasets with variable resolutions.

By analyzing these datasets, we demonstrate that the GBC model can overcome a range of challenges, showcasing its adaptability and robustness in real-world clinical scenarios.

The presented scheme has been evaluated for clean and contaminated regions segmentation on the synthetically degraded images to determine how the GBC model reacts to images corrupted from the distribution on which it has been fitted.

To simulate the accumulation of moisture on the lens of capsule endoscopy during image acquisition, the intensity and density of the fog overlay have been randomly changed. In adding random Gaussian noise to the captured WCE images, the color intensity values of the pixels have been altered based on the Gaussian distribution with zero mean and variable variance in the range of 5 to 50. Defocus blur can occur when the camera lens of the capsule endoscopy fails to accurately focus on the targeted area, resulting in a softened image. A defocused kernel has been convolved with the original image to simulate the effect of an out-of-focus lens. Alias blur and range of radius defocusing determine the strength of defocus blur effect. Motion blur in the capsule endoscopy frames has been simulated by applying different 2-D filters on the baseline images. In the context of capsule endoscopy imaging, random brightness alteration has been algorithmically executed by exponentiating each pixel intensity to a random value within the range of 0.75 to 1.25. The H, S, and V values of the images in the HSV color space have been randomly changed to simulate color aberration in the WCE image acquisition process.

Each baseline image in the curated Kvasir capsule endoscopy dataset has been processed for the six mentioned degradation scenarios. This pipeline ends up in a second set of 12000 synthesized images (2000 images for each scenario).

The GBC model fitting and inference are two distinct processes. The GBC model parameters have been fitted based on the original high-quality images and applied on the intentionally different levels of degraded test images. In general, the spread of responses (std) was greater for the color jitter scenario; on the other hand, responses were most cohesive for the defocus condition. As can be observed in Table 2, the negative effects of color jitter in the segmentation of clean and contaminated regions were mostly evident. This observation was due to the heavy reliance of the GBC model on color information. Similar outcomes have

been observed with changing the illumination conditions and simulating uneven brightness distribution since it has significant effects on the visual color of the small bowel contents because shifts in the light spectrum alter the color pixel intensity values. In the case of fogging, defocus, and motion blur degrading scenarios, the performance of the GBC model is more robust. Sharp edges are not required since the GBC model is resistant to blurriness. Additive Gaussian noise has not been well tolerated by the GBC since this scenario changes the visual color characteristics of the clean and contaminated patterns. Techniques to stabilize the color tonality in the SBCE images and distribute the brightness evenly in the captured images can enhance the clean and contaminated regions in the WCE images.

However, the GBC model has not achieved the same level of accuracy as DCNN structures (U-Net and Pix2Pix) in the segmentation of clean and contaminated regions in the WCE images, but in medical fields such as gastroenterology in which the annotated image data is scarce and expensive to obtain, statistical models can be an optimal choice for the segmentation of clean and contaminated regions in the WCE images.

The GBC model is distinguished from existing deep learning-based segmentation solutions by its minimal data requirements, computational efficiency, and flexibility for multi-class segmentation.

Obtaining large, well-annotated datasets in medical imaging is challenging due to the need for expert knowledge, the labor-intensive nature of annotation, time-consuming, and regulatory/privacy concerns associated with medical data [34]. In contrast to DCNNs [27], which typically require a substantial volume of annotated data for training on new capsule endoscopy databases, our proposed pipeline offers a more efficient alternative by identifying the most informative frames from the pool of unlabeled dataset for the subsequent annotation by the specialist. This targeted selection of frames significantly reduces the burden of labeling efforts compared to annotating all data points, resulting in substantial time and effort savings for endoscopists.

The reduced model construction time of the GBC enables quicker decision-making in clinical settings. For instance, in a busy gastroenterology clinic, the ability to rapidly analyze and classify WCE images translates to shorter patient wait times and more efficient utilization of resources. This accelerated workflow can enhance the overall efficiency of the clinic, allowing healthcare providers to make timely diagnostic decisions and manage patient care more effectively.

Notably, in the model construction phase, would be better to ensure that the selected images adequately represent the visual color characteristics of both clean and contaminated patterns.

Significant reduction in data dependency makes the GBC model particularly applicable in resource-limited settings, where access to extensive labeled datasets and advanced computational resources is often restricted.

Since the GBC model need only a few number of representative annotated images it can be adapted to different datasets, especially for databases with limited annotated data where DCNN-based architectures won't work.

Our lightweight model with very few numbers of learnable parameters (26 parameters) makes it free from large annotated WCE image data. As shown in Table 3, in contrast to U-Net and Pix2Pix architectures with major and minor train and test subset division (80:20 ratio), the constructed dataset has been split with a 1:99 ratio for the GBC construction and evaluation phases.

The DCNN architectures with a huge number of parameters require huge resources, such as a graphical processing unit (GPU) for training. On the other hand, presented low-complexity GBC model with very few numbers of learnable parameters (26 parameters) can

be implemented on a low space memory setup and is well-suited for deployment in environments lacking high-performance GPUs.

The GBC model's low memory and computational requirements, combined with its affordability, make it an ideal choice for clinical environments where both access to high-end computational resources and budgets are limited. It can operate effectively on standard medical computers or even portable devices, minimizing the need for expensive hardware. This ensures broader accessibility and enables healthcare facilities to implement advanced image analysis technologies without imposing significant financial burdens, making it particularly suitable for resource-constrained settings.

From Fig 7, it can be inferred that there is a good correlation in small bowel cleansing evaluation between our experienced gastroenterologist capsule endoscopy reader and the developed GBC model. The available difference in the prediction of clean regions between the GBC model and the gastroenterologist may be due to bias based on clinical experience and the gastroenterologist's notion about clean regions.

## 6. Conclusions

In this manuscript, an automatic clean and contaminated region segmentation scheme has been presented based on the modeling of color intensity values of clean and contaminated pixels. The main concept here is to learn the visual color characteristics of the clean and contaminated regions from the pixel-level knowledge by calculating the prior probability, fitting two different statistical models on the observations, and finally, the Bayes rule has been applied for pixel classification in the evaluation phase to relate the PDF of pixel intensity values given the class to the posterior probability of the class given the data.

Our algorithm could also be helpful from different perspectives. Firstly, it can objectively compare the effect of different bowel preparation paradigms. On the other hand, the reviewing time by gastroenterologists can be significantly reduced by rejecting the contaminated regions. It can assess the capability of different anti-foaming agents in order to decrease gas bubbles and enhance the quality of small bowel visualization.

The developed scheme has been carefully developed in the context of applicability and friendliness. The presented pipeline is particularly valuable in medical scenarios in which the obtaining of annotated image data is expensive and time-consuming. From a large pool of unlabeled data, the most informative fames have been selected to be annotated by the specialist. The expert must mark a few sample images' clean and contaminated regions using a freehand drawing tool. This expert information allows the algorithm to encode the color characteristics of clean and contaminated patterns during the model construction phase process. The gastroenterologist takes only a few minutes to carry out the annotation process.

The GBC model can complement current capsule endoscopy reading software capabilities for small bowel preparation scoring in several ways. First, by automatically segmenting clean and contaminated regions, the model can enhance review efficiency by reducing the time gastroenterologists spend reviewing video frames, allowing them to focus on clinically relevant areas. Second, it can complement existing scoring systems by providing an objective measure of bowel cleanliness, offering segmentation capabilities not available in current capsule endoscopy software. Third, the model could prioritize cases for review based on cleanliness scores, helping gastroenterologists allocate their time more effectively. Finally, it could be used as a training tool for medical students and residents, providing feedback on bowel preparation quality and improving their skills in evaluating WCE images.

The presented framework can be extended to multiclass segmentation tasks just by including the probability models of as many classes as needed, such as different classes of lesions.

While color analysis can provide a powerful tool for the SBVQ evaluation challenge, it is true when some types of pathologies overlap in color with contaminated regions; it is necessary to incorporate other types of features. In such situations, the clean and contaminated region segmentation by the color features must be followed up with texture and shape analysis techniques.

Due to our limited resources, the constructed datasets have been annotated under the supervision of one gastroenterologist. However, it is more desirable to mark the clean and contaminated regions with the aid of many physicians and calculate the overlap between different annotated masks to create the final ground truth.

The different Bayes classifiers differ mainly by the assumption they make regarding the distribution $p(x\,|\,y = k)$. In the current manuscript, the likelihood of the input features has been assumed to be Gaussian. Modeling each class's conditional probability distributions by different PDFs can be carried out as future works. By implementing the presented scheme to a graphical user interface (GUI), it can serve as an educational tool for medical students, residents, and other healthcare professionals.

## Author contributions

**Conceptualization:** Alireza Mehridehnavi, Mohsen Sharifi.

**Data curation:** Vahid Sadeghi, Maryam Behdad, Mohsen Sharifi, Yasaman Sanahmadi, Niloufar Teyfouri.

**Investigation:** Vahid Sadeghi, Maryam Behdad, Mina Omrani.

**Methodology:** Vahid Sadeghi, Alireza Mehridehnavi, Alireza Vard.

**Project administration:** Alireza Mehridehnavi.

**Software:** Vahid Sadeghi, Maryam Behdad.

**Validation:** Vahid Sadeghi, Alireza Mehridehnavi.

**Visualization:** Vahid Sadeghi.

**Writing – original draft:** Vahid Sadeghi, Alireza Mehridehnavi, Maryam Behdad, Mina Omrani.

**Writing – review & editing:** Vahid Sadeghi, Alireza Mehridehnavi, Mina Omrani.

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
