## [Decision Letter · Decision Letter 0]

20 Sep 2024

PONE-D-24-24094Multivariate Gaussian Bayes Classifier with Limited Data for Segmentation of Clean and Contaminated Regions in the Small Bowel Capsule Endoscopy ImagesPLOS ONE

Dear Dr. Mehridehnavi,

Thank you for submitting your manuscript to PLOS ONE. After careful consideration, we feel that it has merit but does not fully meet PLOS ONE’s publication criteria as it currently stands. Therefore, we invite you to submit a revised version of the manuscript that addresses the points raised during the review process.

We look forward to receiving your revised manuscript.

Kind regards,

Xiaohui Zhang

Academic Editor

PLOS ONE

Journal Requirements:

3. In the online submission form, you indicated that [The three different datasets, and codes that support the findings of this manuscript are available from the first author (V.S.), upon reasonable request.]. 

Reviewers' comments:

Reviewer's Responses to Questions

**Comments to the Author**

1. Is the manuscript technically sound, and do the data support the conclusions?

Reviewer #1: Partly

Reviewer #2: Yes

2. Has the statistical analysis been performed appropriately and rigorously? 

Reviewer #1: No

Reviewer #2: Yes

3. Have the authors made all data underlying the findings in their manuscript fully available?

Reviewer #1: Yes

Reviewer #2: No

4. Is the manuscript presented in an intelligible fashion and written in standard English?

Reviewer #1: Yes

Reviewer #2: Yes

5. Review Comments to the Author

Reviewer #1: The manuscript presents a statistical model designed to segment clean and contaminated regions in wireless capsule endoscopy (WCE) images using a multivariate Gaussian Bayes classifier. The main goal is to improve the visualization quality of WCE images, which is often hindered by contaminants like food residue and bubbles. The model uses only 20 manually labeled images for training and applies probabilistic Gaussian distribution models to RGB color pixel intensity values for classification. Its performance was evaluated using the Kvasir, SEE-AI, and CECleanliness datasets. Before publication, I recommend addressing the following points:

1- The manuscript claims the model is novel, cost-effective, and suitable for resource-limited settings. This is highly relevant in medical applications where large annotated datasets are difficult to obtain. However, it would be helpful to elaborate on how this model differs from existing solutions. Can you provide a detailed comparison with other similar models to justify its applicability?

2 - Using various datasets to test the model's robustness and adaptability is commendable. The results show high accuracy and robustness across different datasets. However, a more detailed analysis of the characteristics of each dataset and how they influence the model's performance would be beneficial. What specific challenges did each dataset present, and how did the model overcome them?

3 - The manuscript highlights that while the GBC model may not achieve the highest accuracy compared to deep convolutional neural networks (DCNNs), it is significantly less resource-intensive. This point is critical, especially for clinical settings with limited computational resources. It would strengthen your argument to include detailed resource usage metrics (such as memory and computational time) for both the GBC and DCNN models. How do these differences impact practical applications in a clinical setting?

4 - The process of selecting 20 images for training is not entirely clear. What criteria were used to ensure these images represented both clean and contaminated regions adequately? Providing a more detailed explanation of this selection process and ensuring the diversity of training images would address concerns about the model's generalizability.

5 - The manuscript should explain how the model deals with images that contain both clean and contaminated regions. Is there a specific threshold or rule applied in these cases? This is important because MDs do not drop contaminated images as they still may contain clinically relevant features. How does the model ensure that these features are not missed?

6 - The model's performance varies across different datasets. It would be insightful to discuss the potential reasons for these differences. Have you considered correlating the preparation scores of each image (like SBFVQ or BBPS) with the model's predictions? A subgroup analysis based on lesion type could also provide valuable insights into the model's performance in different clinical scenarios.

7 - Including additional evaluation metrics such as precision, recall, and F1-score, AUROC, would provide a more comprehensive assessment of the model's performance. These metrics can help understand the balance between correctly identifying clean and contaminated regions, and addressing issues with false positives and false negatives.

8 - Discussing the practical applications of this model in a clinical setting would be highly valuable. How do you envision this model being integrated into existing WCE review workflows? Current software already has functions for small bowel preparation scoring. How can this model enhance or complement these existing functions? Providing a roadmap for integration into clinical practice would demonstrate its potential real-world impact.

Reviewer #2: In this work, the authors proposed a multivariate Gaussian Bayes classifier to classify small bowel capsule endoscope images into clean and contaminated regions. Compared to existing neural network-based classifiers, this algorithm only requires 20 manually pixel-labeled images, which is very important in situations with limited labeled images. In addition, this algorithm is also robust to various real-world degradation scenarios, e.g., motion blur, defocus…

Overall, the paper is well-written and has a good structure. The author provides sound reasoning for the design of the algorithm, together with solid evaluations. I would recommend accepting this paper once the authors address the following concerns:

Major concerns:

1. Fig 7: The x and y axis's meaning is unclear. Please define the meaning of clean region. Does it refer to the ratio of clean area in the whole image?

2. Line 339-404: how do the accuracy, DSC, and IOU look like if we apply U-net and Pix2pix algorithms that are trained on the Kvasir dataset to classify SEE-AI and CECleanliness data? If the authors see worse results, this will further help the author to reason that the GBC algorithm can be transferred to different datasets, especially for datasets with limited data where neural network-based algorithms won’t work.

Minor concerns:

1. Line 215: ( = |) and (|) change to p( = |) and (y=c|)

2. Line 342: Assessment ‘of ’ the performance ….

6. PLOS authors have the option to publish the peer review history of their article (what does this mean? ). If published, this will include your full peer review and any attached files.

**Do you want your identity to be public for this peer review?** For information about this choice, including consent withdrawal, please see our Privacy Policy .

Reviewer #1: No

Reviewer #2: No

---

## [Author Response · Author response to Decision Letter 1]

11 Nov 2024

we have respond to each point raised by the academic editor and reviewer(s)in the uploaded 'Response to Reviewers' file.

---

## [Decision Letter · Decision Letter 1]

29 Nov 2024

Multivariate Gaussian Bayes Classifier with Limited Data for Segmentation of Clean and Contaminated Regions in the Small Bowel Capsule Endoscopy Images

PONE-D-24-24094R1

Dear Dr. Mehridehnavi,

We’re pleased to inform you that your manuscript has been judged scientifically suitable for publication and will be formally accepted for publication once it meets all outstanding technical requirements.

Kind regards,

Xiaohui Zhang

Academic Editor

PLOS ONE

Additional Editor Comments (optional):

Reviewers' comments:

Reviewer's Responses to Questions

**Comments to the Author**

1. If the authors have adequately addressed your comments raised in a previous round of review and you feel that this manuscript is now acceptable for publication, you may indicate that here to bypass the “Comments to the Author” section, enter your conflict of interest statement in the “Confidential to Editor” section, and submit your "Accept" recommendation.

Reviewer #1: All comments have been addressed

Reviewer #3: All comments have been addressed

2. Is the manuscript technically sound, and do the data support the conclusions?

Reviewer #1: Yes

Reviewer #3: Yes

3. Has the statistical analysis been performed appropriately and rigorously? 

Reviewer #1: Yes

Reviewer #3: Yes

4. Have the authors made all data underlying the findings in their manuscript fully available?

Reviewer #1: Yes

Reviewer #3: Yes

5. Is the manuscript presented in an intelligible fashion and written in standard English?

Reviewer #1: Yes

Reviewer #3: Yes

6. Review Comments to the Author

Reviewer #1: (No Response)

Reviewer #3: 1. The choice of k=20 in the k-means clustering requires further justification. It would be beneficial if the authors could provide evidence to support this parameter selection.

2. There appears to be a significant discrepancy in the test set sizes between the proposed GBC method (1,980 samples) and the baseline models (U-Net and Pix2Pix, 400 samples each). This discrepancy may lead to an unfair comparison of performance metrics across models. I recommend either aligning the number of test samples or providing a clear rationale for this disparity to ensure a fair and robust comparison.

Minor Comment:

Please ensure the title formatting of all references is consistent. For example, in references 12 and 27, the titles have each word capitalized, while others do not.

7. PLOS authors have the option to publish the peer review history of their article (what does this mean? ). If published, this will include your full peer review and any attached files.

**Do you want your identity to be public for this peer review?** For information about this choice, including consent withdrawal, please see our Privacy Policy .

Reviewer #1: No

Reviewer #3: No

---

## [Editor Report · Acceptance letter]

PONE-D-24-24094R1

PLOS ONE

Dear Dr. Mehridehnavi,

I'm pleased to inform you that your manuscript has been deemed suitable for publication in PLOS ONE. Congratulations! Your manuscript is now being handed over to our production team.

Kind regards,

on behalf of

Dr. Xiaohui Zhang

Academic Editor

PLOS ONE